# Association of Air Pollutants with Incident Chronic Kidney Disease in a Nationally Representative Cohort of Korean Adults

**DOI:** 10.3390/ijerph18073775

**Published:** 2021-04-04

**Authors:** Seo Yun Hwang, Seogsong Jeong, Seulggie Choi, Dong Hyun Kim, Seong Rae Kim, Gyeongsil Lee, Joung Sik Son, Sang Min Park

**Affiliations:** 1School of Health and Environmental Science, Korea University, Seoul 02841, Korea; s_yun@korea.ac.kr; 2Department of Biomedical Sciences, Seoul National University Graduate School, Seoul 03080, Korea; jeongseogsong@snu.ac.kr (S.J.); seulggie@gmail.com (S.C.); 3Department of Medicine, Seoul National University College of Medicine, Seoul 03080, Korea; doraibul@snu.ac.kr (D.H.K.); sungkim20@snu.ac.kr (S.R.K.); 4Department of Family Medicine, Seoul National University Hospital, Seoul 03080, Korea; gespino1.gs@gmail.com; 5Department of Family Medicine, Korea University Guro Hospital, Seoul 08308, Korea; medical114@naver.com

**Keywords:** particulate matter, renal failure, air pollution, ozone, cohort study

## Abstract

(1) Background: There is limited information regarding association between long-term exposure to air pollutants and risk of chronic kidney disease (CKD) (2). Methods: This study acquired data of 164,093 adults aged at least 40 years who were residing in 7 metropolitan cities between 2002 and 2005 from the Korean National Health Insurance Service National Sample Cohort database. CKD risk was evaluated using the multivariate Cox hazards proportional regression. All participants were followed up with until CKD, death, or 31 December 2013, whichever occurred earliest. (3) Results: Among 1,259,461 person-years of follow-up investigation, CKD cases occurred in 1494 participants. Air pollutant exposures including PM_10_, SO_2_, NO_2_, CO, and O_3_ showed no significant association with incident CKD after adjustments for age, sex, household income, area of residence, and the Charlson comorbidity index. The results were consistent in the sensitivity analyses including first and last year annual exposure analyses as well as latent periods-washed-out analyses. (4) Conclusions: Long-term exposure to air pollution is not likely to increase the risk of CKD.

## 1. Introduction

Chronic kidney disease (CKD) is an irreversible and progressive disease with high prevalence globally [1]. In 2017, 697.5 million people were diagnosed with CKD, and 1.2 million people died due to CKD [2]. The overall prevalence of CKD is 13.7% in Korea, which is higher than the worldwide prevalence of 9.1% [3]. To date, old age, diabetes mellitus, hypertension, smoking, obesity, and cardiovascular disease (CVD) were found associated with increased risk of CKD [4,5].

Ambient air pollution, a result of advanced industrialization and urbanization, has emerged as one of the well-known environmental risk factors for public health [6]. In 2012, the World Health Organization (WHO) reported that annual measure of one in eight of entire global deaths is associated with air pollution exposure [7]. Recently, air pollution exposures were newly identified to be a potential risk factor for CKD. However, results of epidemiological studies regarding the association of exposure to air pollutants with risk of CKD demonstrated contradictory results [8,9,10,11]. In addition, most previous studies have focused on association of particulate air pollutants but not the effects of gaseous pollutants, such as ozone (O_3_), with CKD risk [8]. Furthermore, more than half of the studies were cross-sectional studies.

A study from Taiwan demonstrated that exposure to air pollutants including particulate matter (PM_2.5_), sulfur dioxide (SO_2_), nitrogen dioxide (NO_2_), nitrogen oxides (NO_X_), and nitrogen monoxide (NO) was related to increased risk of CKD and end-stage renal disease (ESRD) [12]. In addition, a Korean study that assessed the Korean National Health and Nutrition Examinations Survey (KNHANES) database reported that exposure to PM_10_ and NO_2_ were associated with decreased estimated glomerular filtration rate (eGFR) level, but not CKD [13]. However, there remains further comprehensive confirmation on association of long-term exposure to O_3_, which is a primary oxidant of photochemical smog that has extrapulmonary effects to the blood, central nervous system, spleen, and other organs [14]. CKD develops when attributed by the repetitive renal inflammation that leads to unsuccessful filtering of the blood [1]. Moreover, the kidneys are especially vulnerable to air pollution exposure, considering about 20% of the cardiac output of blood is conveyed to the kidneys, where most environmental toxins would be settled during filtration [6]. Therefore, long-term exposure to air pollution is expected to derive excessive inflammatory factors that contribute to the development of CKD.

Although the adverse effects of air pollution for a number of diseases, including respiratory disease, diabetes, and CVD, are well elucidated in literature, evidence remains regarding the association of air pollutants with other non-communicable diseases, such as CKD, especially in the Korean population [15,16]. Herein, we sought to evaluate the impacts of PM_10_, NO_2_, SO_2_, CO, and O_3_ exposures on risk of incident CKD using a nationally representative cohort from the Korean National Health Insurance Service (NHIS) database.

## 2. Materials and Methods

### 2.1. Study Population

The study population was acquired from the Korean NHIS National Sample Cohort (NHIS-NSC) database. The sample cohort was composed of 2.2% of the eligible participants who received health examination in 2002 [17]. The enrollment rate of the NHIS is about 97%, which includes approximately all population in Korea [18]. Health screening is conducted biannually among all citizens aged 40 years and above and includes a self-reported questionnaire on their behaviors; medical history; lifestyles; sociodemographic characteristics such as age, sex, residence, household income, residential area; and anthropometric measurements, including height, weight, blood pressure, body mass index (BMI), fasting serum glucose (FSG), and total cholesterol (TC) [18]. To protect participants′ privacy according to the Privacy Act, the resident registration number was replaced by a newly allocated eight-digit personal ID [17,18,19]. The Seoul National University Institutional Review Board approved the present study (E-1806-076-951). Informed consent for participation was waived as the database is anonymized under strict confidentiality guidelines by the Korean NHIS.

We obtained data on 193,942 participants aged over 40 years from NHIS-NSC for 11 years (2002–2013). Data of air pollutants were measured from the seven metropolitan cities in Korea (Seoul, Pusan, Daegu, Incheon, Gwangju, Daejeon, and Ulsan). We excluded 27,271 participants who had missing air pollutant information, 854 participants who were diagnosed with CKD before January 2006, and 1724 participants who died before the follow-up investigation. Finally, 164,093 participants were selected for the analyses. As for the sensitivity analysis, we included only those who underwent national health examination with complete information for the adjusted analysis (*n* = 49,380; Figure 1).

### 2.2. Assessment of Air Pollution Exposures

Annual average concentrations of PM_10_, SO_2_, NO_2_, CO, and O_3_ data were obtained from the National Ambient Air Monitoring System (NAMIS) in South Korea. Using the NAMIS, daily air pollution data can be collected from about 200 atmospheric monitoring sites nationwide in the Republic of Korea. The Air Korea database provides data with area codes consisting of administrative residential areas on the basis of the location of each station. All air pollutants were accessible for seven metropolitan cities during 2002–2005. The Korean NHIS database includes residential area codes for each participant. On the basis of residential area codes for each participant, 4-year (2002–2005) exposures of the average air pollutants were calculated, and the participants were stratified according to the quartiles of each air pollutant [20,21].

### 2.3. Follow-Up for CKD Event

We designed a longitudinal study with 8 years follow-up from 1 January 2006 to 31 December 2013. All patients were followed up until CKD, death, or 31 December 2013, whichever occurred earliest. We used hospital admission records and International Classification of Disease, Tenth Revision (ICD-10) for the operational definition of CKD. In the Korean insurance claims database, CKD was defined for those with the diagnostic codes of “N18.x” (N18, N18.1, N18.2, N18.3, N18.4, N18.5, and N18.9), as described in a previous study [22]. Therefore, CKD was considered present when a participant had hospitalization ≥1 time or hospital visit ≥3 times due to an ICD-10 code of N18 [23].

### 2.4. Statistical Analysis

The adjusted hazard ratios (aHRs) and 95% confidence intervals (CIs) for CKD were calculated using the Cox proportional hazards regression model after adjustments for the covariates for each analytic model. Model 1 (main results with overall study population) involved the following covariates: age (categorized to 10-year intervals), sex (men and women), household income (4 quartiles according to the insurance premium), area of residence (seven metropolitan cities), and Charlson comorbidity index (CCI, continuous). Model 2 (sensitivity analysis for exclusion of influences on CKD risk caused by other covariates) additionally adjusted for health screening examination data on the basis of Model 1. In this study, CCI was calculated by diagnoses made between 2002 and 2005. A *p* value of less than 0.05 was considered statistically significant. All data mining and statistical analyses were performed using SAS Enterprise Guide 7.1 (SAS Institute Inc., Cary, NC, USA) and STATA 16.1 (STATA Corp., College Station, TX, USA).

## 3. Results

Table 1 shows descriptive characteristics of the overall study population. There were 164,093 participants with a higher proportion of women (52.2%) compared to men (47.8%). The high proportion was found among participants with 40–49 years of age (43.2%), household income at first (highest) group (33.7%), residence in Seoul (50.8%), and CCI = 0 (40.2%). The mean concentrations of PM_10_, SO_2_, NO_2_, CO, and O_3_ were 61.7 μg/m^3^, 0.006 ppm, 0.031 ppm, 0.620 ppm, and 0.018 ppm, respectively.

Appendix A presents the descriptive characteristics of the participants who underwent health examination (*n* = 49,380) according to annual exposure to PM_10_, which accounts for 30.1% of the total population. The mean BMI, systolic blood pressure (SBP), FSG, and TC levels were 23.9 kg/m^2^, 125.7 mmHg, 98.0 mg/dL, and 199.4 mg/dL, respectively. Approximately 15% of the participants drank 2–3 times a month, and 24.7% of the participants drank more than once a week. The portion of participants with no regular physical activity was 47.4% (*n* = 23,405). In addition, the percentages of former and current smokers were 5.3% (*n* = 2623) and 23.0% (*n* = 11,379), respectively. Furthermore, the proportion of the participants with CCI = 0 was higher (40.2% versus 34.6%) compared to those who received health examinations.

The numbers of CKD event, person-year, incidence rate, and aHR with 95% CI for CKD according to the concentration of PM_10,_ SO_2_, NO_2_, CO, and O_3_ are shown in Table 2. After adjustments for age, sex, insurance premium, area of residence, and CCI, the highest PM_10_ (aHR, 1.06; 95% CI, 0.87–1.29), SO_2_ (aHR, 1.11; 95% CI, 0.92–1.35), NO_2_ (aHR, 1.01; 95% CI, 0.80–1.28), CO (1.01; 95% CI, 0.85–1.19), and O_3_ (aHR, 1.15; 95% CI, 0.86–1.54) revealed no significant difference compared to the corresponding lowest groups, respectively.

For sensitivity analyses, we first tested whether the main findings are consistent using the annual exposure for each the earliest (2002) and latest (2005) year within the participant enrollment period (Appendix A). The results were similar to the main findings except for O_3_, which showed increased CKD risk in the second (aHR, 1.21; 95% CI, 1.02–1.43) and third (aHR, 1.16; 95% CI, 1.00–1.34) quartiles compared to the first quartile in the analysis conducted using the annual exposure in 2005.

Considering the nature of CKD that develops in a chronic manner, we washed out one, three, and five years of latent periods to exclude cases with developing CKD due to other causes (Appendix A). In accordance with the main results, no significant association was detected in all air pollutant exposures. Furthermore, we enlarged the exposure period from four years to seven years to confirm whether longer period of exposure contributes to the development of CKD, which also indicated no significant association between air pollutant exposures and the risk of CKD (Appendix A).

Table 3 shows sensitivity analyses that further adjusted for health screening results, including factors previously identified as risk factors for the development of CKD and lifestyle behaviors among participants who underwent health examination (*n* = 49,380). All aHRs were around 1.00 and no significant difference was found in five air pollutant exposures. Taken together, air pollution seems not to be associated with incident CKD in various manners.

Figure 2 shows the association of the air pollutant exposures with incident CKD in subgroups stratified according to age, sex, household income, and CCI. According to the interaction analyses, CKD risk among concentration groups of PM_10,_ SO_2_, NO_2_, CO, and O_3_ were significantly affected by age (*P*_interaction_ < 0.001), while O_3_ was additionally affected by CCI (*P*_interaction_ = 0.002). In addition, second and fourth quartiles of the participants aged below 60 years revealed significantly increased CKD risk compared to the first quartile, suggesting that a relatively younger population may be more susceptible to CO exposure in terms of CKD risk. Similar results were also found in the CO concentration groups among participants with upper half of the household income.

## 4. Discussion

In summary, our data indicate that there is no notable increase in CKD risk after long-term exposure to PM_10_, SO_2_, NO_2_, CO, and O_3_. To the best of our knowledge, this is the first longitudinal cohort study in Korea that assessed data from the NHIS to evaluate the association of air pollutants with incident CKD. Despite evaluation of the additional exposure variable, our data suggested that there is no significant association of air pollutant exposures with risk of CKD.

Considering recent findings regarding effects of air pollution in incident cancer and respiratory diseases, such as chronic obstructive pulmonary disease (COPD) and asthma, it is clear that air pollutants highly contribute to the development of chronic disease [24]. In terms of kidney diseases, exposure to environmental pollutants including heavy metals, agricultural and industrial chemicals, and biogenic toxins are notable risk factors [6]. However, the relationship between air pollution and CKD has not yet been fully elucidated; thus, our study has strengths regarding the exposure period as well as the follow-up investigation that was carried out for eight years, which allowed evaluation of long-term exposures to the air pollutants.

Similar to our results, a Korean cross-sectional study reported that exposures to PM_10_ and NO_2_ are significantly associated with decreased eGFR levels but not CKD [13]. It is meaningful because our cohort study supports this result with enlarged study population (*n* = 164,093) and addition of O_3_ as an exposure variable. In a recent study from Taiwan, one-year exposures to PM_2.5_ and NO_2_ were associated with lower eGFR, higher CKD prevalence, and higher risk of CKD progression among elders [25]. However, considering the analytic population (*n* = 8497) residing in Taipei City, their result may be underpowered. In addition, their adjusted analyses included age, sex, BMI, education, smoking, alcohol drinking, hypertension, and diabetes but not comorbidities, such as CCI. In another study from Taiwan, acidic gas air pollution and particulate matter were found associated with higher risk of CKD and ESRD after adjustments for age, sex, income, and urbanization level [12]. Although CKD often arises in patients with comorbidities, such as hypertension, diabetes, and coronary disease, both studies from Taiwan had insufficient consideration of comorbidities [26].

Up to date, the underlying correlation between air pollution and CKD has not yet been completely clarified. However, there are some common characteristics between the mechanism of incident CKD and the effects of air pollution. A possible hypothesis is that exposure to air pollutants leads to increased levels of inflammatory biomarkers, such as C-reactive protein, interleukin 6 and 8, and TNF-α [26]. These pro-inflammatory cytokines promote the development of CKD, and some inflammatory biological markers were found increased after exposure to air pollution [27,28]. Likewise, most previous studies focused on pathogenic mechanisms such as oxidative stress, systemic inflammation, and damage to distant organs [29]. An experimental study of rats showed that concentrated O_3_ exposures can bring oxidative stress and inflammation [30]. Despite this potential mechanism, our data indicated there is no independent association of air pollutants, including O_3_, with CKD. Thus, it is important to have a further investigation on the relationship between environmental pollutants and genetic factors which may affect disease susceptibility [6]. However, there requires a caution in interpretation of the data that no significant increase in risk is limited to CKD only since other diseases, such as stroke, ischemic heart disease, COPD, asthma, and osteoporosis, were previously reported to be significantly affected by exposure to the air pollution [31,32,33].

Compared to participants aged under 60 years within the first quartile of CO, those in the second and fourth CO quartile groups were at higher risk for CKD. In addition, CO concentration groups among participants with upper half of the household income resulted in higher CKD risk in the highest quartile group. According to previous studies that explored the association of CO exposure with CKD risk, a study with a mean age of under 60 years (58.3 years) demonstrated an adjusted odds ratio of 1.19 (95% CI, 1.02–1.39; *p* = 0.030) per IQR increase in the exposure to CO, which is in accordance with our data [34]. Another potential reason for significantly increased CKD risk according to the exposure of CO among participants aged under 60 years is that these participants are less affected by unfavorable impact of age on CKD risk compared to those aged 60 or more. Participants with relatively older age had higher baseline CKD risk regardless exposure to CO, which may have weakened potential unfavorable effects of CO. Therefore, we still call for future studies with larger study population to better evaluate association of the exposure to CO with CKD risk.

Our study had several limitations. First, we used air pollutant concentration data according to the area of residential district code; thus, the migration of participants during the follow-up investigation could not be accounted for. Second, indoor air quality exposure, participant’s personal activity, and use of masks could not be considered in the present study due to data unavailability. Third, factors that may contribute to the development of CKD, such as creatinine, high-density lipoprotein particle, and low-density lipoprotein particle may not have been comprehensively considered due to limited information. Additional adjustments for other significant factors using an information-comprehensive database would be required in the future. Fourth, our study population consisted of Eastern population; further verification by a Western population-base is required for general application of the data. Fifth, the causes of acute kidney injury (AKI), such as chemotherapy, surgery, and contrast studies, could not be considered due to data unavailability regarding the retrospective nature of the study. Sixth, the stage of CKD at diagnosis in follow-up periods was not evaluated in the present study; thus, we call for future studies to confirm whether exposures to air pollutants are associated with the stage of CKD at diagnosis. Lastly, our study used a retrospective cohort study design. We could not actively and unlimitedly select variables of interest but selected from a preexisting database [35]. Despite the underlying limitations listed above, our study has benefits that may reduce those biases supported by a large sample size and good quality control of the database.

## 5. Conclusions

In conclusion, we found that long-term exposure to air pollutants, including PM_10_, NO_2_, SO_2_, CO, and O_3_, is not independently associated of incident CKD in Korean adult population aged at least 40. Higher exposure to CO significantly increased CKD risk in age <60 years and upper half of the household income subgroups, while all other subgroups showed consistency with the main finding that CKD is not significantly affected by exposure to air pollutants. In addition, we suggest that it is unnecessary for healthy behaviors, such as outdoor physical activity, to be disturbed due to the air pollution in terms of CKD risk.

## Figures and Tables

**Figure 1 ijerph-18-03775-f001:**
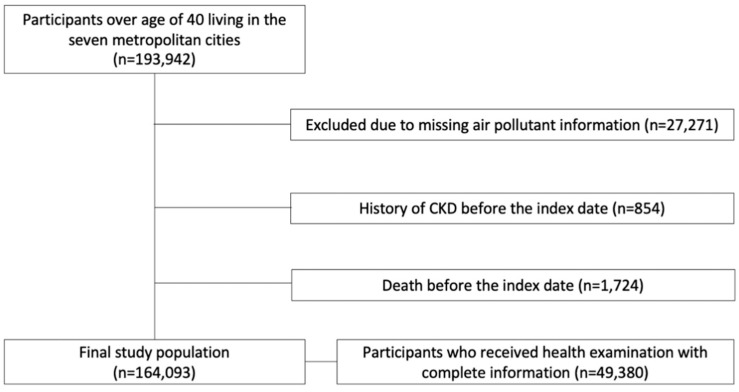
Study population inclusion flowchart.

**Figure 2 ijerph-18-03775-f002:**
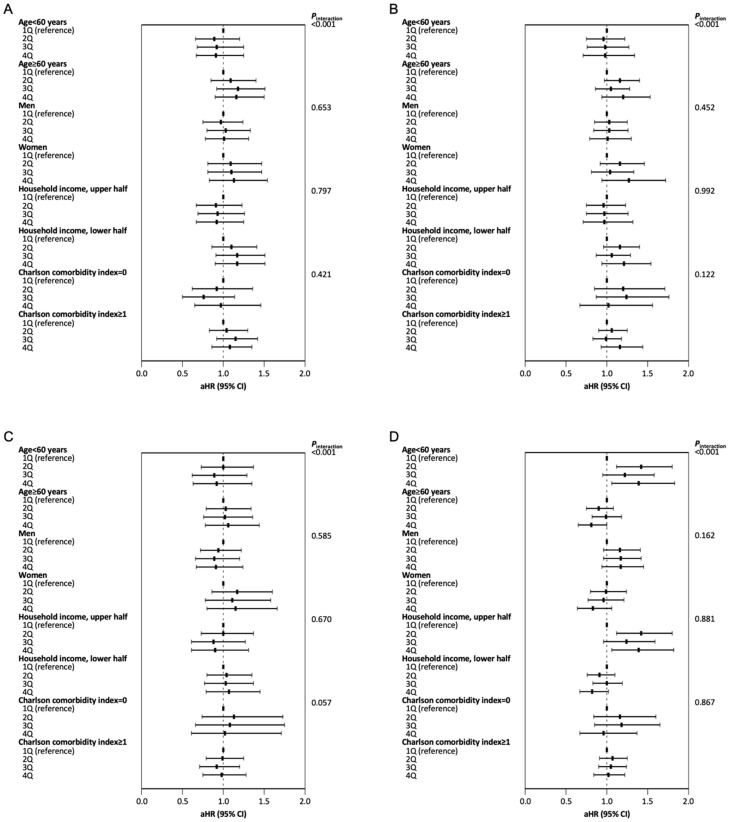
Subgroup analysis on association of the air pollutants with incident chronic kidney disease (CKD). (**A**) Subgroup analysis on association of PM_10_ with CKD. (**B**) Subgroup analysis on association of SO_2_ with CKD. (**C**) Subgroup analysis on association of NO_2_ with CKD. (**D**) Subgroup analysis on association of CO with CKD. (**E**) Subgroup analysis on association of O_3_ with CKD.

**Table 1 ijerph-18-03775-t001:** Descriptive characteristics of the overall study population.

Characteristic	Participants (*n* = 164,093)
Age, years	
40–49	70,944 (43.2)
50–59	47,167 (28.7)
60–69	28,541 (17.4)
≥70	17,441 (10.6)
Sex	
Men	78,459 (47.8)
Women	85,634 (52.2)
Insurance premium	
1st (highest)	55,333 (33.7)
2nd	47,487 (28.9)
3rd	34,194 (20.8)
4th (lowest)	27,079 (16.5)
Charlson comorbidity index	
0	65,958 (40.2)
1	41,662 (25.4)
≥2	56,473 (34.4)
Residential area (7 Metropolitan cities)	
Seoul	83,360 (50.8)
Pusan	30,718 (18.7)
Daegu	10,055 (6.1)
Incheon	17,945 (10.9)
Gwangju	8943 (5.5)
Daejeon	5460 (3.3)
Ulsan	7612 (4.6)
Concentration of PM_10_, μg/m^3^, mean (SD)	61.7 (7.8)
Concentration of SO_2_, ppm, mean (SD)	0.006 (0.002)
Concentration of NO_2_, ppm, mean (SD)	0.031 (0.007)
Concentration of CO, ppm, mean (SD)	0.620 (0.100)
Concentration of O_3_, ppm, mean (SD)	0.018 (0.004)

Data are *n* (%) unless indicated otherwise. Acronyms: PM, particulate matter; SD, standard deviation.

**Table 2 ijerph-18-03775-t002:** Association of air pollutants with incident chronic kidney disease among Korean adults aged at least 40 years.

Air Pollutant	Quartiles of Air Pollutants in Annual Average				*P* _trend_
	First Quartile	Second Quartile	Third Quartile	Forth Quartile	
PM_10_, μg/m^3^, range	37.3–58.1	58.1–62.9	63.4–65.3	65.4–81.5	
Event (%)	331 (0.80)	373 (0.89)	385 (0.96)	405 (0.99)	
Person-year	316,668	322,350	306,710	313,733	
Incidence rate	1.05	1.16	1.26	1.29	
aHR (95% CI)	1.00 (reference)	1.02 (0.84–1.24)	1.06 (0.88–1.29)	1.06 (0.87–1.29)	0.468
SO_2_, ppm, range	0.0035–0.0045	0.0045–0.0054	0.0054–0.0062	0.0064–0.0120	
Event (%)	392 (0.89)	387 (0.96)	364 (0.91)	351 (0.88)	
Person-year	337,152	310,163	306,301	305,844	
Incidence rate	1.16	1.25	1.19	1.15	
aHR (95% CI)	1.00 (reference)	1.08 (0.93–1.25)	1.03 (0.88–1.21)	1.11 (0.92–1.35)	0.395
NO_2_, ppm, range	0.015–0.027	0.027–0.032	0.032–0.037	0.037–0.043	
Event (%)	323 (0.77)	416 (0.94)	359 (0.94)	396 (1.00)	
Person-year	322,744	338,983	293,401	304,333	
Incidence rate	1.00	1.23	1.22	1.30	
aHR (95% CI)	1.00 (reference)	1.03 (0.84–1.26)	0.97 (0.78–1.22)	1.01 (0.80–1.28)	0.886
CO, ppm, range	0.28–0.58	0.58–0.64	0.66–0.68	0.69–0.84	
Event (%)	387 (0.86)	382 (0.95)	404 (0.97)	321 (0.86)	
Person-year	346,019	310,543	318,646	284,252	
Incidence rate	1.12	1.23	1.27	1.13	
aHR (95% CI)	1.00 (reference)	1.08 (0.94–1.25)	1.08 (0.93–1.25)	1.01 (0.85–1.19)	0.777
O_3_, ppm, range	0.012–0.014	0.014–0.018	0.018–0.021	0.021–0.030	
Event (%)	381 (0.92)	465 (1.03)	328 (0.85)	320 (0.82)	
Person-year	320,567	346,623	294,789	297,482	
Incidence rate	1.19	1.34	1.11	1.08	
aHR (95% CI)	1.00 (reference)	1.13 (0.98–1.29)	1.18 (0.93–1.50)	1.15 (0.86–1.54)	0.179

aHR calculated by Cox proportional hazards regression. Adjusted for age, sex, insurance premium, area of residence, and Charlson comorbidity index. Acronyms: PM, particulate matter; aHR, adjusted hazard ratio; CI, confidence interval; SO_2_, sulfur dioxide; NO_2_, nitrogen dioxide; CO, carbon monoxide; O_3_, ozone.

**Table 3 ijerph-18-03775-t003:** Association of the air pollutants with incident chronic kidney disease among the participants who received health examination.

Air Pollutant	Quartiles of Air Pollutants in Annual Average				*P* _trend_
	First Quartile	Second Quartile	Third Quartile	Forth Quartile	
PM_10_, μg/m^3^	1.00 (reference)	1.02 (0.84–1.24)	1.06 (0.88–1.29)	1.06 (0.87–1.29)	0.468
SO_2_, ppm	1.00 (reference)	1.08 (0.93–1.25)	1.03 (0.88–1.21)	1.11 (0.92–1.35)	0.395
NO_2_, ppm	1.00 (reference)	1.03 (0.84–1.26)	0.97 (0.78–1.22)	1.01 (0.80–1.28)	0.886
CO, ppm	1.00 (reference)	1.08 (0.94–1.25)	1.08 (0.93–1.25)	1.01 (0.85–1.19)	0.777
O_3_, ppm	1.00 (reference)	1.18 (0.85–1.65)	0.74 (0.41–1.31)	0.56 (0.29–1.10)	0.322

Data are adjusted to hazard ratio (95% confidence interval) and calculated by Cox proportional hazards regression after adjustments for age, sex, insurance premium, smoking, alcohol consumption, physical activity, systolic blood pressure, fasting serum glucose, total cholesterol, area of residence, and Charlson comorbidity index.

## Data Availability

To access the database used in the present study, one should submit the security memorandum and pledge to the Institutional Review Board of Korean NHIS. After the approval, data are provided with anonymized personally identifiable information. Any other researchers can access the data in the same manner. Contact information for data accessibility is listed as follows: Tel.: +82-337-362-432, Website: https://nhiss.nhis.or.kr, accessed on 4 March 2021.

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
