# Peer review of "Association of Air Pollutants with Incident Chronic Kidney Disease in a Nationally Representative Cohort of Korean Adults"

_ijerph, 2021, doi:10.3390/ijerph18073775_

Round 1

Reviewer 1 Report

Seo Yun Hwang et al. described the “Association of Air Pollutants with Incident Chronic Kidney 2 Disease in a Nationally Representative Cohort of Korean Adults”. They used the Korean National Health Insurance Service National Sample Cohort database and determined the risk for incident CKD among 164,093 adults aged at least 40 years from 7 metropolitan cities between 2002 and 2005. Of note, air pollutant exposures, including PM10, SO2, NO2, CO, and O3, showed no significant association with incident CKD after adjustments for age, sex, household income, area of residence, and the Charlson comorbidity index. Sensitivity analyses were done and found similar results. This is interesting and still some questions to be solved.

Major:

Q1: As authors exclude history of CKD before the index date, how authors determine such population should be explained. E.g. which parameters used, how long and which stages of CKD, etc.

Q2: Since acute kidney injury (AKI) will have many causes including chemotherapy, surgery, contrast studies, etc., how authors exclude such clinical settings? Authors use ICD-10 code of N18 with hospitalization≥1 time; this is not sufficient for CKD, and more likely to be AKI or AKD.

Q3: People from different residential areas (7 Metropolitan cities) included in the study, how authors determine the regional variation, since most people differ in their residential places and working areas, and how to interpret influence of different air pollution status in these people.

Q4: Discussion should include details on the findings in Figure 2; where different groups of air pollutant exposures with CKD risks, especially age<60yrs.

Q3: In CKD determination, did authors evaluate stages of CKD? Any difference in development of early and late CKD events?

Minor:

Q4: English should be reviewed again professionally from a native English speaker.

Reviewer 2 Report

I consider that this is an interesting manuscript that should be accepted for publication in Intern J. Environ. Res. Public Health, after minor revision. The manuscript focus on the association of air pollutant namly PM10, SO2, No2, CO and O3 and their possible relationship with the chronic kidney incidence in a nationally representative cohort of Korean adults considering available data from a cohort database, concluding that long-term exposure to the air pollution focused in the referred pollutant is not likely related with the enhancement of the risk of CKD. Some considerations should be taken account by authors previously to the final acceptance namely:

  • Although authors stated (lines 78-79) that “ The consent for publication was waived as the database is anonymized under strict confidentiality guidelines, however when a human study is performed is previously mandatory to inform and therefore obtain the signed consent of participants in the study.
  • Table 1 legend is incorrect; it refers to information to authors of the journal. Please write the appropriate one.
  • Information of Figure 2 is not legible. In this sense, higher letter size of figures 2A,B,C,D,E should be supplied in the revised version. I would recommend to supply this information in two whole pages in order to facilitate the reading of researchers interested in this study.

Round 2

Reviewer 1 Report

Agree with the authors' response and no more comments.